# Digestive Constraints of *Arthrospira platensis* in Poultry and Swine Feeding

**DOI:** 10.3390/foods11192984

**Published:** 2022-09-24

**Authors:** Maria P. Spínola, Mónica M. Costa, José A. M. Prates

**Affiliations:** 1CIISA—Centro de Investigação Interdisciplinar em Sanidade Animal, Faculdade de Medicina Veterinária, Universidade de Lisboa, Av. da Universidade Técnica, 1300-477 Lisboa, Portugal; 2Associate Laboratory for Animal and Veterinary Sciences (AL4AnimalS), Faculdade de Medicina Veterinária, Universidade de Lisboa, Av. da Universidade Técnica, 1300-477 Lisboa, Portugal

**Keywords:** microalgae, *Arthrospira platensis*, bioavailability, bio-accessibility, digestibility, broiler, pig

## Abstract

Microalgae have emerged as novel sources for monogastric animals’ diets since they are rich in many nutrients, including proteins. *Arthrospira platensis* is particularly rich in proteins (up to 76% of dry matter), lipids, minerals and pigments. However, its rigid peptidoglycan cell wall interferes with the digestibility, bio-accessibility and bioavailability of nutrients for monogastric animals. The aim of the present study was to evaluate the digestibility, bio-accessibility, bioavailability and protein quality of nutrients from *A. platensis* for poultry and swine feeding, searching all the studies available in PubMed, Web of Science, Scopus and Google Scholar in June 2022 concerning this subject. Overall, digestibility values of *A. platensis* proteins or amino acids varying from 66.1 to 68.7% for poultry (microalgae at 1% feed) and from 75.4 to 80.6% for swine (10% feed) have been reported. Therefore, the extraction of microalgae components using mechanical or non-mechanical pre-treatments is required to promote cell disruption and improve digestibility and bio-accessibility. Although *A. platensis* is a promising feedstuff to support future needs, it is important to perform more investigation concerning digestibility, dietary inclusion level and possible treatments to disrupt microalga cell walls and increase bioavailability of nutrients.

## 1. Introduction

The growth in human population makes the search for alternative sources for animal feeding, especially those rich in proteins, imperative [1]. Since the 1950s, a lot of research has been performed in an attempt to find alternative protein sources, in anticipation of the estimated lack of protein supply in the forthcoming years [2].

Microalgae have emerged as promising sources with high protein content and biological value [2]. They are aquatic microscopic organisms that can have high growth rates and are usually divided into four different groups, according to pigmentation, life cycle and cell structure: diatoms (*Bacillariophyceae*), green algae (*Chlorophyceae*), blue-green algae (*Cyanophyceae*) and golden algae (*Chrysophyceae*) [3]. The majority is autotrophic with photosynthetic activity, but some of them are heterotrophic [4]. Microalgae do not require arable land for cultivation or compete for a limited space, which makes them sustainable sources [5,6,7]. They can grow in open oceans, rocky shores, freshwater habitats (rivers, lakes, ditches and ponds) and deserts [8,9]. In addition, they could present high concentrations of proteins, which are comparable to or higher than those of conventional protein sources used in animal nutrition, as well as carbohydrates and lipids, even though their chemical composition varies depending on species, culturing and growth conditions (i.e., temperature, light and salinity) [10].

*Arthrospira platensis*, also known as Spirulina, is the most cultured microalga worldwide, since, from 1950 to 2019, 99.6% of 56,456 tonnes of total microalgae production corresponded to Spirulina [11]. This alga is an autotrophic blue-green microalga, with a prokaryotic structure typical of a *Cyanobacteria* and photosynthetic activity [2]. It preferentially grows in fresh water, in alkaline lakes, but also in saltwater, which affects alga mineral content [10,12,13]. The microalga biomass is rich in proteins and bioactive compounds [4]. Although there is a lack of information concerning the use of *A. platensis* in swine diets [14], the microalga has been studied as a supplement in poultry diets, and the results demonstrate that it acts as a natural colour enhancer for meat [15] and eggs [16]. However, *A. platensis* has a poor digested peptidoglycan cell wall, especially for monogastric animals such as poultry and swine [17]. Therefore, it is essential to properly process this microalga before feeding it to these animals in order to avoid impairment of nutrients´ bioavailability and bio-accessibility [4]. In addition, an optimization of microalgae production towards a reduction of its economic and environmental impact is necessary, taking advantage of the current technological advances associated with algae cultivation, such as the use of wastewater and industrial carbon dioxide. This is expected to enhance the sustainability and diminish the costs of algae production [18].

Regarding the safety of *A. platensis* for feed and food consumption, this microalga is generally recognised as safe by the European Food Safety Authority, and *A. platensis* products are authorized for consumption under specific conditions in the EU, including their use as food supplements [19]. Indeed, *A. platensis* is one of the microalgae used as food in the EU prior to 1997 and, therefore, is not considered as a novel foodstuff [20,21]. A previous study by Martín-Girela et al. [22] emphasized the safety of *A. platensis* for dietary supplementation, showing that several contaminants from freshwater (i.e., hormones, pharmaceutical and personal care products, flame retardants and biocides) were present below the detection limits in the microalga. However, Manali, et al. [23] detected microcystins in Spirulina used as food supplements for fish, although those cyanotoxins were not found in human food, and advised standard cultivation and production conditions to guarantee the quality of microalgae. Therefore, the maximum limit of microcystins, as well as their toxicity, in *A. platensis* should be defined using realistic testing methodologies and recognizing the difference between the level (1 μg/L) and toxicity of microcystins in drinking water and in cyanobacterial supplements. Indeed, microalgae have several antioxidants, such as pigments, vitamin E and polysaccharides, which can reduce the toxicity of these substances [24].

The objective of the present study was to evaluate the digestibility, bio-accessibility and bioavailability of nutrients from *A. platensis* biomass and the respective improvement challenges, covering the literature available in PubMed (NCBI, Bethesda, MD, USA), Web of Science (Clarivate Analytics, Philadelphia, PA, USA), Scopus (Elsevier B.V., Amsterdam, The Netherlands) and Google Scholar (Google LLC, Mountain View, CA, USA). The literature search was performed in June 2022 using the keywords “*Arthrospira platensis*”, “Spirulina”, “bioavailability”, “bio-accessibility”, “digestibility”, “poultry”, “swine” and “pig”.

## 2. Nutritional Composition of *Arthrospira platensis*

*A. platensis* has a rich nutritional composition (Table 1), although variable and largely dependent on production conditions [10]. The protein content is considerably high (up to 76% of dry matter, DM) with a good quality, since it contains all the essential amino acids [4,8,25], which makes this microalga one of the top choices as alternative to traditional protein sources [26]. The most predominant amino acids are glutamic and aspartic acids, leucine, alanine and arginine [10], with lower amounts of methionine and cysteine [26,27]. For poultry, lysine and methionine are essential for limiting amino acids [28]. In addition, for swine, lysine is also the first-limiting amino acid, but the proportion of tryptophan, threonine and methionine is also relevant [29]. The microalgae also contain considerable amounts of lysine (average of 7.8% of total amino acids), which is a limiting amino acid for monogastric animals (Table 1).

In addition, *A. platensis* contains a considerable amount of carbohydrates (up to 22.6% DM) [4], although a previous study reported lower values (6.5% DM) [8]. The variability of carbohydrate contents depends on the culture period, with a proportional increase in carbohydrates, mainly glucose, with microalga cultivation time. This phenomenon occurs in detriment of other alga components, such as proteins [10]. The main carbohydrate fraction include soluble carbohydrates [8,27], but insoluble polysaccharides are important components of recalcitrant microalga cell wall. These are composed of a multilayer structure of glucan and peptidoglycan polymers covered by acidic polysaccharides [30].

*A. platensis* is also rich in vitamins, such as vitamin E (tocopherols) and some vitamins of B complex, and antioxidant pigments including carotenoids (e.g., β-carotene), allophycocyanin, C-phycocyanin, chlorophyll *a* and xantophylls, such as zeaxanthin [10,31]. The latter were shown to confer a yellow [32] or red [16] colour to chicken eggs and yellow colour to meat [15], which modifies consumers’ perception of these animal products [25]. *A. platensis* has biliproteins, especially C-phycocyanin, which is a bioactive compound acting as antioxidant, regulator of immunity and, thus, protector of the organism against disease [26]. The mineral content of *A. platensis* can vary between 3 and 11% DM [25], and calcium, magnesium, potassium and sodium are amongst the most predominant minerals [27]. Total lipids are present in variable amounts in *A. platensis* (1.8–16% DM), but this microalga could be a good source of some polyunsaturated fatty acids (PUFA), such as linolenic acid (18:3n-6) [25,27].

Overall, the nutritional quality of *A. platensis*, particularly in what concerns the protein fraction, makes it a good alternative to conventional feedstuffs (e.g., soybean meal). However, there are still difficulties related to microalga production and bioavailability of algal nutrients [4,27], which could be overcome, respectively, by technological advances associated with more sustainable algae cultivation [18] and by dietary supplementation with feed enzymes [33] or algae pre-treatments.

## 3. Digestibility, Bio-Accessibility and Bioavailability of Nutrients

It is important to differentiate the concepts of digestibility, bio-accessibility and bioavailability of nutrients. In fact, digestibility is the fraction of feed components that is transformed by digestion into possibly accessible matter and then absorbed by the animal. It is calculated as the amount of nutrient consumed minus that retained in the faeces [17,52]. The digestibility can be divided into ileal and faecal. The determination of the first implies digesta collection at the ileal junction from cannulated animals or slaughtered small animals, such as poultry [53], whereas, for the last, faecal collection and analysis are executed. In turn, bio-accessibility is the nutrient amount or fraction that is released in the gastrointestinal tract and is available for absorption through the intestinal epithelia. It is evaluated by in vitro procedures that simulate gastric and intestinal digestion [17,52]. Finally, bioavailability is the fraction of ingested nutrient that is absorbed, reaches the circulatory system and becomes available at the site of action. The process to render a nutrient bioavailable includes gastrointestinal digestion, absorption, metabolism and tissue distribution [17,52,54]. Digestibility is commonly but incorrectly referred to as a synonym of bioavailability. Indeed, the latter can only be assessed by performing animal growth bioassays using live animals [53].

Bioavailability is a common concept in animal nutrition, especially concerning energy metabolism. It is known that, when feeding animals nutrients, they are not totally digested and metabolized by the animals, and two processes are critical, digestion and metabolism [55]. For instance, the protein bioavailability can be measured by digestibility, biological value (BV) or net protein utilization (NPU) [2]. A more accurate method for the determination of amino acid digestibility involves collecting ileal digesta or using cecectomized animals instead of analysing faeces because of the influence of microbial protein produced either in caeca, in poultry, or in large intestine, in swine [56,57,58]. However, there are some difficulties with the determination of amino acid digestibility from collected faeces, because the last portion of the intestinal microbiota can modify some undigested amino acids. Therefore, it is of upmost importance to clarify nitrogen metabolism in intestine. The degradation of nitrogenous compounds, the synthesis of microbial protein and the balance of these two modifies the amount of amino acids in excreta compared to those found in the ileum. The degradation implies deamination of amino acids and ammonia formation, which can be absorbed, but it is usually excreted via urine. If degradation is superior to synthesis, the amino acids in excreta will decrease and amino acid digestibility will be overestimated. On the other hand, if synthesis is superior to degradation, there will be an increase in amino acid in excreta and a sub-estimated amino acid digestibility [58]. This aspect is important in swine [28] and in poultry, but in the latter, the influence of microbial protein is unclear [58], and faeces and urine are mixed due to excretion via common cloaca. Therefore, it is more correct to analyse metabolizable instead of digestible protein values [58,59]. In addition, Parsons et al. [60] observed that the percentage of amino acids present in poultry excreta (25%) with microbial origin was lower than that found for swine (50%) [61]. However, the ileal cannulation method is a quite difficult procedure due to the viscosity of this intestinal portion, which makes it very hard to keep the cannula in the right conditions [28]. Thus, the collection of ileal content from euthanized animals previously fed with a diet containing an indigestible marker is an easier method to evaluate amino acid digestibility [7]. In addition, Gamboa-Delgado et al. [35] proposed one method to estimate nutrient assimilation, which consisted in determining the isotopic signatures in ingredients and animal tissue. Nitrogen and carbon isotope ratios were considered natural biomarkers. However, this method was only described for shrimp and not for poultry or swine.

Several factors modify digestibility and bioavailability of alga nutrients, such as unbalanced amino acids in the diet, source of protein (animal or plant), feed processing methods (heating, roasting or extrusion), antagonism, protein quality, and structure, level and composition of fibre [53,56]. The type of grain, application of fertilizer and environmental conditions can affect amino acids´ digestibility [56]. Feed processing methods can disrupt antinutritional factors present in microalgae, but overheating in particular can make amino acids unavailable and unbalanced. For instance, in poultry, the antagonism between lysine and arginine is common, where lysine stimulates the catabolism of arginine and, thus, decreases its intestinal absorption [53,62].

The rigid polysaccharide-rich cell wall present in microalgae (about 10% of dry matter) can inhibit the bioavailability of algal nutrients because the digestive system of monogastric animals cannot digest microalgae cells and the bio-accessibility of intracellular metabolites is low [2,6,63]. Thus, some studies have reported the importance of effective treatments to disrupt the cell wall and make the nutrients more bio-accessible [2]. Extraction of algae components and mechanical (mechanical forces, liquid-shear forces, energy transfer through waves and currents or heat) or non-mechanical (cell lysis with chemical agents, enzymes or osmotic chock) pre-treatments of microalgae can possibly promote cell disruption and improve digestibility and bio-accessibility of nutrients [17]. High-pressure homogenization, high-speed homogenization, bead milling and ultrasonication are the four methods most utilized on laboratory scale for microalgae cell disruption [17].

Considering non-mechanical methods, the use of pepsin and pancreatin to simulate gastrointestinal digestion and Carbohydrate-Active enZymes (CAZymes) to decompose fibre components has proven to be effective in disrupting *A. platensis* cell wall and increasing the bio-accessibility of algal nutrients. The assessment of in vitro digestibility of *A. platensis* using a combination of pepsin and pancreatin showed high organic matter, crude protein and carbohydrate digestibility for the algal biomass, with values of 86, 81 and 79%, respectively [64]. Noticeably, the crude protein digestibility value was comparable with conventional protein sources (up to 78%) [64]. Misurcova et al. [36] reported even high percentages of *A. platensis* dry weight digestibility using pepsin (74.1–89.6%) and pancreatin (82.9–97.5%), with only slight differences when combining the two enzymatic extracts (85.6–94.3%). Furthermore, Kose et al. [65] observed 64% of hydrolysis yield of microalga biomass degraded with pancreatin. In what concerns the use of CAZymes, few studies reported their efficiency in disrupting *A. platensis* cell wall. Coelho et al. [33] showed that an enzyme mixture with lysozyme and α-amylase was able to disrupt the microalga cell wall, after 16 h of in vitro incubation under controlled experimental conditions [66], and release valuable nutrients to the supernatant, including total protein (142 mg/g microalgae) (*p* = 0.018), n-6 PUFA (e.g., 18:2n-6) (2.8%) (*p* = 0.007), monounsaturated fatty acids (2.47%) (*p* = 0.049) and chlorophyll *a* (0.066 mg/g microalgae) (*p* = 0.025). Overall, the enzymatic mixture enhanced the nutritional composition of *A. platensis* supernatant. This was accompanied by a decrease in proteins in *A. platensis* residue, although there was still a significant increase in algal nutritive and bioactive compounds in this fraction such as chlorophyll a and some PUFA (i.e., 18:2n-6, 18:3n-3, 22:2n-6). The nutritional composition of the control and enzyme-treated residue is presented in Table 2.

No digestibility assays were performed in this experiment [33], but one in vivo study described that the combination of 10% of *A. platensis* with lysozyme fed to piglets compromised the apparent total tract digestibility (ATTD) of protein. A subsequent proteomic study on *longissimus lumborum* muscle reported an increase in structural muscle protein synthesis with higher energy requirements for piglets fed CAZyme-treated microalgae [67]. Although carbohydrases are able to degrade *A. platensis* cell wall, this causes a release of proteins that are resistant to endogenous peptidases and enhance digesta viscosity. Therefore, additional hydrolysis of algal proteins is necessary to increase their bio-accessibility and digestibility for monogastric animals.

Regarding mechanical methods, an adaptation of bead milling using a stirred ball mill before in vitro digestion assays was shown to improve in vitro protein digestibility of non-disrupted *A. platensis* from 74 to 78% [39]. Indeed, the bead milling technique can enhance the bioavailability of protein and fatty acids from microalgae biomass because of its ability to disrupt algae cell walls [6]. Table 3 shows the summary of main effects of in vitro pre-treatments on hydrolysis and digestibility of *A. platensis*.

## 4. Methods to Evaluate Bio-accessibility and Bioavailability of Nutrients and Protein Quality

The protein quality depends on the amino acid profile of the diet, the content of essential and limiting amino acids and the digestibility and physiological utilization of amino acids after digestion, which are very relevant for monogastric animals [53].

Protein bioavailability and protein quality can be assessed by different methods including in vivo trials or in vivo assays. Performing a digestibility assay is a good method for determination of amino acids bioavailability because of its simplicity, since it is based on faecal or excreta analysis, and it can be applied to a high number of animals without implying sacrifice. The BV measures the proportion of protein that is absorbed and incorporated into proteins of animals’ bodies, which shows the percentage of absorbed protein retained in the body and how quickly the digested protein can become available for synthesis. The NPU indicates the ratio of amino acids converted to proteins in the body to the amino acids supplied in the diet. It can range between 0, indicating none of the dietary protein was retained in the body, and 1, indicating 100% utilization of dietary nitrogen as protein. The Protein Efficiency Ratio is the ratio between body weight gain and protein consumed, assuming that all the protein is used for growth [53].

According to Bryan and Classen [68], some in vitro techniques, which are easier to perform and less expensive than in vivo trials, can evaluate protein quality in poultry. There are chemical in vitro methods (protein solubility index with potassium hydroxide or protein dispersibility index), the pH-Stat/Drop method and closed enzymatic methods. The last method can be a pepsin, pancreatin or multi-enzymatic assay. Some factors can influence protein digestion such as enzymatic specificity and activity, protein structure and forms, anti-nutritive agents or even the test samples. Although in vitro assays do not truly replicate in vivo conditions, they give an estimation of protein bio-accessibility and are good alternatives to in vivo digestibility assays.

Neumann et al. [69] measured the protein quality of piglet and growing pig diets using N balance assays. The diets consisted of 21% or 13% of Spirulina meal for piglets or pigs, respectively, with amino acids supplementation. In this study, productive protein value (PPV) and NPU were used to evaluate the complex dietary protein quality concerning the process of digestion and post-absorption utilization. PPV and NPU depend on the actual level of protein intake. The results showed that the dietary supplementation with amino acids enhanced protein quality, and supplementation with histidine in Spirulina meal diet led to a superior protein quality.

Moreover, the production of bioactive compounds is a different process that allows the isolation of proteins and peptides with functional functions from microalgae. This methodology increases the bio-accessibility of valuable compounds and includes a first step of protein extraction using various mechanical and non-mechanical methods as mentioned before [17], followed by protein enrichment and purification through dialysis, ultrafiltration, and ion-exchange chromatography. For protein characterization, SDS-PAGE, mass spectrometry or colorimetric assays can be applied. Then, protein hydrolysis is performed by enzymatic or chemical hydrolysis or microbial fermentation. When peptides are released from the protein, amino acid composition, hydrophobicity and molecular weight influence their bioactivity [70].

## 5. Bioavailability of *Arthrospira platensis* Nutrients for Poultry

One of the most important nutrients for animal nutrition, which also constitutes a large proportion of *A. platensis* biomass, is protein. Therefore, it is important to clarify the bioavailability of amino acids, particularly of lysine and methionine, which are essential and limiting amino acids for poultry [28]. Additionally, glycine is an essential amino acid to chickens [53]. Noticeably, the amount of methionine in the microalga is usually low (average of 4.1% of total amino acids), which might necessitate an increase in dietary incorporation levels of *A. platensis*. The protein digestibility was found to be lower at the initial stages of poultry growth and to increase afterwards, within the first two weeks of life. Therefore, it is important that the animals have access, in the first two weeks, to highly digestible ingredients and that their requirements are set high for digestible amino acids [28]. The growth assay is one of the methods to determine amino acid bioavailability and is considered a gold standard method, although it is not a digestibility assay. For instance, a diet can be supplemented with two or more levels of a tested amino acid, followed by evaluation of growth performance parameters. This procedure estimates the digestion, absorption and utilization of each amino acid for protein synthesis, but it is time-consuming, and many variables can interfere with the results [28]. Another option is to perform a digestibility assay, which includes the use of cecectomized animals, collection of faeces and urine separately and analysis of faeces for its amino acid content [28,71]. This type of assay was applied to broiler chickens fed a fish meal diet [71]. The authors showed that cecum has an important role in protein digestion for poultry, since cecectomized animals could not absorb digesta containing amino acids, in contrast to intact broilers, which indicates that the amino acids are probably absorbed and digested in cecum. However, to the best of our knowledge, there is no study describing a digestibility assay using cecectomized poultry fed microalgae. In fact, the cecostomy is easy to perform and has few complications but demands surgery, thereby requiring additional expertise [28].

An in vivo digestibility assay in broilers, where an indigestible marker (acid-insoluble ash) was applied followed by collection of ileal contents, showed that feeding *A. platensis* as an ingredient (20% feed) showed an increase in animal body weight with enhanced amino acid digestibility [7]. The authors reported standardized ileal amino acid digestibility coefficients for essential (0.80) and non-essential (0.78) amino acids [7].

Studies about *A. platensis* digestibility and nutrient assimilation are scarce [72]. This might be not only due to the difficulties of evaluating nutrient digestibility in poultry but also due to the controversial results obtained on animal growth performance when feeding with *A. platensis*, which are dependent on the alga level in feed. Park et al. [31] reported that the inclusion of 1% of *A. platensis* increased the ATTD of dry matter from 67.9 to 71.1% and nitrogen from 66.1 to 68.7%, which improved broiler growth, probably due to a better nutrient absorption. Conversely, Pestana et al. [73] showed that a basal diet containing 15% Spirulina supplemented with a recombinant lysozyme reduced broiler growth performance. This was likely due to an increase in intestinal viscosity capable of compromising the accessibility of digestive enzymes to their substrates and, thus, nutrient digestibility. The authors suggested that the CAZyme disrupted alga cell wall and released proteins from microalga biomass, which were resistant to endogenous birds’ peptidases and formed a viscous matrix trapping valuable nutrients [72]. This viscosity was probably due to a complex phenomenon of gelation, including unfolding and aggregation steps, that algal proteins suffer when denatured at high temperature (>60 °C) [74]. Table 4 presents a summary of main effects on nutrient digestibility of dietary inclusion of *A. platensis* in in vivo trials on poultry.

## 6. Bioavailability of *Arthrospira platensis* Nutrients for Swine

In swine, weaning is a critical life phase, and microalgae, due to their bioactive compounds acting as prebiotics, have been studied for their potential health benefits. These sources can boost intestinal health and avoid severe diarrhoea that delays piglet development, thus reducing the need for antibiotics in the post-weaning period [3,75]. Therefore, feeding swine *A. platensis* may represent a solution for maintaining good digestive function after weaning [75]. In addition, the microalga biomass can contain up to 11.3% lysine out of the total amount of amino acids (Table 1), which is the first-limiting amino acid for swine [29]. For newly weaned piglets’ feeding, in addition to lysine itself, the ratio between each amino acid (threonine, total sulphur amino acids, tryptophan, valine, isoleucine) and lysine is also relevant [76,77]. Particularly, the tryptophan requirement in the diet for 21-to-45-day-old piglets was found to be 1% of the dietary protein (0.613% for *A. platensis*). Thus, feeding *A. platensis* would provide such a level of this essential amino acid (1.4% of total amino acids) (Table 1). For growing-finishing pigs, the list of amino acids required is long and includes all the essential (with the exception of phenylalanine) and two conditionally essential amino acids (arginine and cysteine). For gestation sows, the list is shorter, consisting of lysine, threonine, tryptophan, isoleucine, valine, methionine, and cysteine [78]. When the microalga does not provide the recommended amount of amino acids, it is necessary to supplement them in the diet. Although *A. platensis* seems to be a good alternative for conventional protein sources in swine diets [79], most studies used this microalga as a supplement and not as a functional ingredient [80]. Indeed, low doses of *A. platensis* were shown to improve nutrient digestibility, in contrast to high doses. For instance, feeding *A. platensis* at 1% to piglets significantly increased the gross energy digestibility coefficient (90.9 to 92.8%), and also tended to enhance DM (91.3 to 93.0%), organic matter (92.4 to 93.8%) and neutral detergent fibre digestibility (77.3 to 81.0%), which was suggested to be caused by an effect of microalga on intestinal mucosa architecture with an increase in villus height and villus height: crypt depth ratio in jejunum [75].

In addition, high levels (10% feed) of *A. platensis* were reported to affect total tract apparent digestibility for all the nutrients, with the exception of neutral detergent fibre. It decreased total tract apparent digestibility of crude protein (80.6 to 75.4%) in post-weaning piglets, even though the dietary supplementation with a recombinant lysozyme increased crude fat (55.6 to 62.8%) and acid detergent fibre (23.0 to 37.3%) digestibility relative to animals fed a corn- and soy-based diet or solely the algae, respectively [80]. Nevertheless, Altmann et al. [40] and Neumann et al. [69] demonstrated that 9.5 and 13%, respectively, of *A. platensis* incorporated in growing-finishing pigs´ diets with amino acid supplementation, mostly lysine, had no negative effects on animal growth or pork quality. Although the former source did not analyse the effects on nutrient digestibility of such high levels of alga, Neumann et al. [69] showed no significant differences in apparent nitrogen digestibility. Table 5 is a summary of main effects on nutrient digestibility of dietary inclusion of *A. platensis* in in vivo trials on swine.

## 7. Conclusions and Future Perspectives

Overall, *A. platensis* is a promising novel ingredient to support the future needs in poultry and swine production. This microalga has a high protein value and several nutritional and bioactive compounds, such as fatty acids, polysaccharides, minerals and pigments, that can stimulate animal growth performance and health. However, there are some limitations on the use of microalgae as a feedstuff at a large scale associated with the cultivation process (high production costs, mostly related to drying and conditioning of algae biomass) and the bio-accessibility and bioavailability of microalgae nutrients. Although scarcely reported, the values of protein or amino acids digestibility of A. platensis range between 66.1 and 68.7% for poultry and from 75.4 to 80.6% for swine. Even though these values of protein digestibility are interesting, this microalga has a recalcitrant cell wall composed of polysaccharides that are mostly indigestible for poultry and swine, which reduces nutrient digestibility.

Therefore, for an increase in microalga nutrient bio-accessibility, it is necessary to enhance algal biomass digestibility by disrupting the cell wall using, for instance, mechanical (e.g., bead milling) or non-mechanical (e.g., feed enzymes) pre-treatments. However, depending on the A. platensis inclusion level, additional procedures might be necessary to increase nutrient bioavailability, such as the use of exogenous proteases that would degrade the gel-forming matrix of proteins released from digested microalga biomass. Further studies could help to explore the aspects concerning the bio-accessibility and bioavailability of A. platensis nutrients for monogastric animals.

## Figures and Tables

**Table 1 foods-11-02984-t001:** Detailed nutritional composition of *Arthrospira platensis* (all values are expressed on a dry matter basis; hyphenated values are ranges based on several studies, and mean values are within parenthesis).

Nutritional Composition	*Arthrospira platensis* ^1^
Crude protein (%)	26.0–75.6 (61.3)
Amino acid profile (% total amino acids)	
Alanine	7.8–10.2 (8.6)
Arginine	7.0–14.9 (9.9)
Aspartic acid	9.8–17.6 (12.4)
Cystine	0.7–2.3 (1.3)
Glutamic acid	14.1–22.5 (17.7)
Glycine	4.9–6.9 (5.6)
Histidine	1.6–3.3 (2.4)
Isoleucine	5.3–12.3 (8.1)
Leucine	9.0–22.4 (14.3)
Lysine	4.4–11.3 (7.8)
Methionine	2.4–6.3 (4.1)
Phenylalanine	4.4–10.5 (7.6)
Proline	3.7
Serine	5.2–7.1 (5.9)
Threonine	5.0–10.5 (7.4)
Tryptophan	0.85–2.0 (1.4)
Tyrosine	4.0–9.4 (6.5)
Valine	6.2–13.9 (9.1)
Ash (%)	6.1–19.8 (9.0)
Macrominerals	
Calcium (g/kg)	0.2–9.2 (4.0)
Magnesium (g/kg)	1.6–4.1 (3.4)
Phosphorus (mg/kg)	1.5–13.9 (8.4)
Potassium (g/kg)	13.7–27.8 (17.6)
Sodium (g/kg)	4.8–27.0 (14.3)
Microminerals	
Copper (mg/kg)	1.2–5.1 (3.2)
Iron (g/kg)	0.2–1.1 (0.8)
Manganese (mg/kg)	39.2–54.0 (44.3)
Selenium (mg/kg)	1.1–38.0 (13.5)
Zinc (mg/kg)	25.3–31.1 (27.6)
Crude carbohydrates (%)	4.0–44.8 (17.8)
Non-fibre carbohydrates	7.9–20.9 (15.7)
Crude fibre (%)	0.1–5.0 (2.6)
Acid detergent lignin	0.1–3.2 (1.6)
Acid neutral fibre	0.3–18.3 (6.4)
Neutral detergent fibre	0.2–32.6 (11.0)
Crude fat (%)	0.9–14.2 (5.8)
Fatty acid profile (% total fatty acids)	
16:0	26.6–71.2 (40.1)
16:1n-7	1.8–13.5 (5.3)
18:0	0.7–8.8 (2.9)
18:1n-9	1.5–35.7 (12.0)
18:2n-6	7.9–28.2 (16.8)
18:3n-3	0.6–3.0 (1.0)
18:3n-6	2.7–28.6 (9.2)
20:0	0.02–15.7 (8.1)
20:4n-6	0.3–0.4 (0.4)
20:5n-3	0.1–2.9 (1.3)
22:6n-3	2.3–3.5 (3.0)
Pigments (mg/kg)	
Total carotenoids	743–2230 (1450)
Total chlorophylls	1324–3635 (2455)
β-carotene	248–1497 (872)
Vitamins (mg/kg)	
A	1.0
B1	6.7–44.9 (31.2)
B12	1.9–17.5 (9.7)
B2	38.0–56.1 (47.1)
B3	130–202 (161)
B5	14.8
B6	6.5–12.1 (8.5)
B8	0.1
B9	0.5–500 (250)
K	13.9
α-Tocopherol	26.2
β-Tocopherol	1.0
γ-Tocopherol	1.0

^1^ Supporting sources: Martins, et al. [4], Ljubic, et al. [5], Holman and Malau-Aduli [25], Batista, et al. [34], Gamboa-Delgado, et al. [35], Misurcova, et al. [36], Mohammadi, et al. [37], Tibbetts, et al. [38], Wild, et al. [39], Altmann, et al. [40], Holman, et al. [41], Macias-Sancho, et al. [42], Grinstead, et al. [43], Radhakrishnan, et al. [44], Dalle Zotte, et al. [45], Shabana, et al. [46], Aouir, et al. [47], Tokuşoglu and üUnal [48], Alghonaim, et al. [49], Bennamoun, et al. [50], Bensehaila, et al. [51].

**Table 2 foods-11-02984-t002:** Total protein, pigments and fatty acids (mg/g microalgae) and fatty acid profile (% total fatty acids) of *A. platensis* residue after treatment with Carbohydrate-Active enZymes [33].

Nutritional Composition	*A. platensis* Control	*A. platensis* with Enzyme Treatment
Total protein	669	586
Total carotenois	3.04	2.73
Total chlorophylls	6.46	8.71
Total fatty acids	46.7	41.8
16:0	41.3	41.3
16:1n-7	1.51	1.51
18:0	3.10	2.74
18:1n-9	2.43	2.23
18:2n-6	18.4	18.8
18:3n-3	0.090	0.106
18:3n-6	24.7	24.6
20:0	0.202	0.224
22:2n-6	0.066	0.103

Control: *A. platensis* suspension incubated with PBS, Enzyme treatment: *A. platensis* suspension incubated with mix of enzymes.

**Table 3 foods-11-02984-t003:** Summary of main effects of in vitro pre-treatments on hydrolysis and digestibility of *A. platensis* biomass.

Pre-Treatment of *A. platensis*	Main Effects	References
Combination of pepsin and pancreatin	Increase in organic matter, crude protein and carbohydrate digestibility	Niccolai, et al. [64]
Pepsin or pancreatin	Increase in dry weight digestibility (89.6% with pepsin, 97.5% with pancreatin and 94.3% with combination of the two enzymes)	Misurcova, et al. [36]
Pancreatin	64% of hydrolysis yield of microalga biomass	Kose, et al. [65]
Enzyme mixture with lysozyme and α-amilase	Release of n-6 PUFA, monounsaturated fatty acids and chlorophyll *a*	Coelho, et al. [33]
Bead milling before in vitro digestion assays	Improvement of protein digestibility (74% for non- disrupted cells vs. 78% for disrupted cells)	Wild, et al. [39]

**Table 4 foods-11-02984-t004:** Summary of main effects on nutrient digestibility of dietary inclusion of *A. platensis* in in vivo trials on poultry.

Animals (Age/Initial Body Weight)	Inclusion Level in Feed and Duration of Trial	Main Effects	References
1-day-old male broilers	20% with an indigestible marker for 10 days	Enhanced amino acid digestibility	Tavernari, et al. [7]
1-day-old male broilers weighing 41.5 ± 0.5 kg	1% for 35 days	Increased apparent total tract digestibility of dry matter and nitrogen	Park, et al. [31]
1-day-old male broilers	15% supplemented with a recombinant lysozyme for 14 days	Nutrient digestibility was not analysed	Pestana, et al. [73]

**Table 5 foods-11-02984-t005:** Summary of main effects on nutrient digestibility of dietary inclusion of *A. platensis* in in vivo trials on swine.

Animals (Age/Initial Body Weight)	Inclusion Level in Feed and Duration of Trial	Main Effects	References
Piglets weaned at 28 days weighing 9.1 ± 1.1 kg	1% for 14 days	Increased gross energy digestibility coefficient Tended to enhance dry matter, organic matter and neutral detergent fibre digestibility	Furbeyre, et al. [75]
Piglets, weaned at 28 days weighing 12.0 ± 0.89kg	10% for 28 days	Increased total tract apparent digestibility of crude fat and acid detergent fibre (lysozyme supplementation) Decreased total tract apparent digestibility of crude protein (with or without lysozyme)	Martins, et al. [80]
9 week-old growing-finishing pigs weighing 22 ± 1.6 kg	9.5% with amino acid supplementation, mostly lysine, for 35 to 49 days	Digestibility was not analysed	Altmann, et al. [40]
Growing-finishing pigs weighing 60 kg	13% with amino acid suplementation, mostly lysine. for 10 days	No significant effect on apparent nitrogen digestibility	Neumann, et al. [69]

## Data Availability

The data presented in this study are available on request from the corresponding author.

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
