# Peer review of "Digestive Constraints of Arthrospira platensis in Poultry and Swine Feeding"

_foods, 2022, doi:10.3390/foods11192984_

Round 1

Reviewer 1 Report

I think the review can be improved by reviewing other methods to assess protein quality.

The proposal is aimed at improving the bioaccessibility and bioavailability  of nutrients from A. platensis microalgae,  through the hydrolysis of cell walls, which are indigestible for poultry and swine

The review of this research is oriented to the use of enzymes and processes ( particle size reduction) to achieve the main objective. Which is interesting and relevant.

This topic has been working on legumes and until now it has not been possible to obtain a high efficiency in the hydrolysis of the cell wall compounds. I believe that the key is to obtain an efficient enzyme cocktail to achieve this goal. The review does not address the efficiency of Carbohydrate-Active enzymes, highlighting the improvement of protein digestibility by the use of proteases.

What is new is the application of carbohydrate-active enzymes. But this mention should be checked if it works under standardized experimental conditions.

This review is assignable to a native speaking reviewer, and totally clear and easy to read

The conclusions are based on the evidence and arguments presented by other authors. But I think the review can be improved by including the materials and methods section, with emphasis on the methods used to evaluate the bioaccessibility and bioavailability of nutrients and protein quality of A. platensis.

In table 1, the nutritional content of A. platensis is presented, but it would highlight the importance of the revision, if to this were added nutritional content after enzyme treatment.

Table 1 would be more explicit if another column was created, showing the nutrient content of the enzyme-treated  A. platensis. It would be advisable to review and include other methods to assess the bioavailability of nutrients from A. platensis.

Author Response

Reviewer 1

I think the review can be improved by reviewing other methods to assess protein quality.

Reply: Thanks for your suggestion. A section was added to the revised document “4. Methods to Evaluate Bioaccessibility and Bioavailability of Nutrients and Protein Quality” (lines 302-349, page 8 and 9).

The proposal is aimed at improving the bioaccessibility and bioavailability of nutrients from A. platensis microalgae, through the hydrolysis of cell walls, which are indigestible for poultry and swine. The review of this research is oriented to the use of enzymes and processes (particle size reduction) to achieve the main objective. Which is interesting and relevant. This topic has been working on legumes and until now it has not been possible to obtain a high efficiency in the hydrolysis of the cell wall compounds. I believe that the key is to obtain an efficient enzyme cocktail to achieve this goal. The review does not address the efficiency of Carbohydrate-Active enzymes, highlighting the improvement of protein digestibility by the use of proteases. What is new is the application of carbohydrate-active enzymes. But this mention should be checked if it works under standardized experimental conditions.

Reply: Thanks for your comments and suggestions. In the revised document the section CAZymes was enhanced (lines 263-283, page 7). This review was assignable to a native speaking reviewer, and totally clear and easy to read.

The conclusions are based on the evidence and arguments presented by other authors. But I think the review can be improved by including the materials and methods section, with emphasis on the methods used to evaluate the bioaccessibility and bioavailability of nutrients and protein quality of A. platensis.

Reply: Thanks for your suggestion. A section was added to the revised document “4. Methods to Evaluate Bioaccessibility and Bioavailability of Nutrients and Protein Quality” (lines 302-349, page 8 and 9).

Table 1 would be more explicit if another column was created, showing the nutrient content of the enzyme-treated A. platensis. It would be advisable to review and include other methods to assess the bioavailability of nutrients from A. platensis.

Reply: Thanks for your suggestion. A new table (section 3, table 2, page 7) with nutritional composition of A. platensis (control and enzyme-treated algae) is in the revised document.

Reviewer 2 Report

This review summarizes the most important findings regarding the dietary inclusion of Arthrospira platensis and its digestibility in monogastrics such as poultry and swine. However, the authors could have made a greater effort to get more out of the research. Also, more graphs such as a microalgae leakage or something like that should be made available. A summary table of the studies cited, among other things. In general the review is poor and does not contain recent studies (only two articles 2022). Not to mention that there is one recently published manuscript: Altmann, B.A.; Rosenau, S. Spirulina as animal feed: opportunities and challenges. Foods 2022, 11, 965, which is very similar.

In addition, I think the authors should look for another journal more oriented to animal nutrition.

The abstract itself is poor and should have more details to attract the attention of the readers: Line 15- how much protein?...microalgae in general are a good source of lipids, mineral salts and pigments.

Line 20. Include ranges of dietary inclusion of microalgae.

Author Response

This review summarizes the most important findings regarding the dietary inclusion of Arthrospira platensis and its digestibility in monogastrics such as poultry and swine. However, the authors could have made a greater effort to get more out of the research. Also, more graphs such as a microalgae leakage or something like that should be made available. A summary table of the studies cited, among other things. In general the review is poor and does not contain recent studies (only two articles 2022). Not to mention that there is one recently published manuscript: Altmann, B.A.; Rosenau, S. Spirulina as animal feed: opportunities and challenges. Foods 2022, 11, 965, which is very similar.

Reply: Thanks for your comments and suggestion. Three different Tables summarizing the studies cited are in the revised document. We also acknowledged the reviewer´s concern about the lack of articles published in 2022. The first version had four papers of 2022, two in the text and two in Table 1 (Demarco, 2022; Mohammadi, 2022; Alghonaim, 2022; Altmann,2022) and now the revised document has six (Demarco,2022 [17]; Mohammadi, 2022 [33]; Alghonaim, 2022 [45]; Altmann, 2022 [71], Lopes, 2022 [61] and Lucakova, 2022 [64]). In addition, a summary table of the studies cited was created (Table 3,4 and 5, page 8,10 and 11-12, respectively) Concerning the similarity with Altmann paper, our work is a review about the bioaccessibility, digestibility and bioavailability of A. platensis nutrients, mostly from protein, for poultry and swine, whereas Altmann et al. paper is a general short communication about the effects on system productivity and product quality of feeding monogastric animals with Spirulina.

In addition, I think the authors should look for another journal more oriented to animal nutrition.

Reply: Thank you for your suggestion but we think that this review fits very well in the scope of this journal.

The abstract itself is poor and should have more details to attract the attention of the readers: Line 15- how much protein?... microalgae in general are a good source of lipids, mineral salts and pigments.

Line 20. Include ranges of dietary inclusion of microalgae.

Reply: Thank you for your suggestions. The abstract was modified according to the suggestions.

Reviewer 3 Report

First of all there are some important publication which are not cited, e.g.

A Prominent Superfood: Spirulina platensis
Written By
Nilay Seyidoglu, Sevda Inan and Cenk Aydin

What about safety of applied microalgae? Are they allowed/registered in Europe? (EFSA publications on safety of a new additives)

Lines 149-171 needs to be revised. In faeces there is lot of endogenous amino acids which caused underestimation of apparent digestibility, not overestimation as authors have written. Moreover in poultry faeces there are amino acids excreted via kidneys. To improve the discussion on amino acids, authors should go back to literature from 1970's and 1980's.

Amino acid composition of protein should also be discussed (protein biological value). Generally authors do not provide animal cathegory used in cited literature. For example in early weaning pigs the requirements for some amino acids (e.g. tryptophan) are not met in algae protein.

Author Response

First of all there are some important publication which are not cited, e.g. A Prominent Superfood: Spirulina platensisWritten ByNilay Seyidoglu, Sevda Inan and Cenk Aydin

Reply: Thank you for your suggestions. Relevant information of this paper was added to the revised document (lines 110 to 113 and 132-134, page 3)

What about safety of applied microalgae? Are they allowed/registered in Europe? (EFSA publications on safety of a new additives)

Reply: Thank you for your suggestions. A paragraph concerning this subject was added to the revised document (lines 67-94).

Lines 149-171 needs to be revised. In faeces there is lot of endogenous amino acids which caused underestimation of apparent digestibility, not overestimation as authors have written. Moreover, in poultry faeces there are amino acids excreted via kidneys. To improve the discussion on amino acids, authors should go back to literature from 1970's and 1980's.

Reply: Thank you for your suggestions. The discussion of this subject was modified to the revised document (section 3, lines 196-200 and 203-218pages 5 and 6).

Amino acid composition of protein should also be discussed (protein biological value). Generally, authors do not provide animal category used in cited literature. For example, in early weaning pigs the requirements for some amino acids (e.g. tryptophan) are not met in algae protein.

Reply: Thank you for your suggestions. The discussion about protein biological value is now presented in the revised document (section 4, lines 299-346). The availability and requirement of algal amino acids was also discussed in section 3, lines 203-218; section 5, lines 352-357 and section 6, lines 407-421).

Round 2

Reviewer 1 Report

Incorporation of the suggested changes in the manuscript is verified

Reviewer 2 Report

I reviewed the article and believe that the authors have made sufficient changes to improve the manuscript.

Reviewer 3 Report

Add EFSA publications.